

# Characterization of a self-sustained, water-based condensation particle counter for aircraft cruising pressure level operation

Patrick Weber[1,4], Oliver F. Bischof[1,2], Benedikt Fischer[1], Marcel Berg[1], Susanne Hering[3], Steven Spielman[3], Gregory Lewis[3], Andreas Petzold[1,4] and Ulrich Bundke[1]

[1]Forschungszentrum Jülich GmbH, Institute of Energy and Climate Research 8 – Troposphere (IEK-8), Jülich, Germany
[2]TSI GmbH, Particle Instruments, Aachen, Germany
[3]Aerosol Dynamics Inc, Berkeley, CA, 94710, USA
[4]Institute for Atmospheric and Environmental Research, University of Wuppertal, 42119 Wuppertal, Germany

*Correspondence to*: Patrick Weber (p.weber@fz-juelich.de) and Ulrich Bundke (u.bundke@fz-juelich.de)

**Abstract.** Aerosol particle number concentration measurements are a crucial part of aerosol research. Vertical profile measurements and high-altitude/low-pressure performance of the respective instruments become more and more important for remote sensing validation and as a key tool for the observation of climate variables. This study tests the new, commercially available, water condensation particle counter (MAGIC 210-LP) for the deployment at aircraft cruising pressure levels, that the European research infrastructure IAGOS (In-service Aircraft for a Global Observing System) is aiming for by

operating measurement instrumentation on board passenger aircraft. We conducted a series of laboratory experiments for conditions, which simulate passenger aircraft flight altitude operations. We demonstrate that this type of water condensation particle counter shows excellent agreement with a butanol-based instrument used in parallel, and a Faraday cup aerosol electrometer serving as the reference instrument. Experiments were performed with test aerosols ammonium sulphate and fresh combustion soot at pressure levels ranging from 700 hPa down to 200 hPa. For soluble particles like

ammonium sulphate, the 50% detection efficiency cut-off diameter ($D_{50}$) was around 5 nm and did not differ significantly for all performed experiments. For non-soluble fresh soot particles, the $D_{50}$ cut-off diameter did not differ significantly for particle sizes around 10 nm, whereas the $D_{90}$ cut-off diameter increased from 19 nm at 700 hPa to 37 nm at 200 hPa. The overall counting efficiency for particles larger than 40 nm reaches 100% for working pressures of 200 hPa and higher.

## 1 Introduction

Condensation Particle Counters (CPC) experienced a rise in use in recent years, driven by the increasing awareness of the adverse effects that particles can have on climate change. air quality, and public health, and all their interrelations (Von Schneidemesser et al., 2015). Specifically, the monitoring of atmospheric aerosol on ground (McMurry, 2000) as well as on airborne platforms (Petzold, 2012), the measurement of exhaust aerosol from various sources (Giechaskiel et al., 2009; Petzold et al., 2011; Bischof et al., 2019),  indoor aerosol (Salimifard et al., 2020), and airborne viruses in the still ongoing

pandemic (Somsen et al., 2020) are current key applications of condensation particles counters.

A condensation particle counter (CPC) measures the aerosol particle number concentration by activating nanometre-size particles in a supersaturated environment and further growing them to detectable droplets in the small micrometre size range. Single particles are subsequently detected and counted utilizing a photodetector measuring the intensity of the scattered radiation of a laser beam. John Aitken is known for his early experiments in which he started counting particles

which had grown in an expansion chamber due to the supersaturation of water vapour (Aitken, 1888).

In general terms, the measurement principle of a CPC can be broken down into three steps: saturation by which a supersaturated vapour of a working fluid is formed inside the CPC, condensation by which vapour condenses on the particle,



and detection, by which the enlarged particles scatter light when passing through a laser which is then counted by a photodiode e.g. {Bischof, 2022;Hinds, 1999 }.

Today, mainly three working fluids are in use, n-butyl alcohol (or n-butanol), water and perfluoro-tributylamine (FC43, Fluorinert). For all working fluids, detection efficiency experiments have been conducted over a certain operation pressure range (e.g.,(Brock et al., 2000; Bundke et al., 2015; Gallar et al., 2006; Hermann et al., 2007) which demonstrated the applicability of each working fluid for low-pressure operation CPCs. It should be noted that the use of both butanol and FC43 is limited by the fact that (1) butanol is a flammable liquid and (2) FC43 is a strong greenhouse gas, whereas water has the

advantage to avoid health and safety concerns of butanol. Disadvantageously, water has a three times higher mass diffusion coefficient (Hering et al., 2005; Mei et al., 2021) which increases the consumption of the working fluid during operation.

The European research infrastructure IAGOS (Petzold et al., 2015; www.iagos.org) aims to cover all essential climate variables of the atmosphere, including aerosol particles (Bojinski et al., 2014) employing regular and global-scale measurements conducted on board of a fleet of passenger aircraft equipped with automated scientific instrumentation. The IAGOS aerosol

instrument using butanol-based CPC is described in detail by Bundke et al. (2015) and provided the first results during the observation of the Raikoke volcanic ash plume by IAGOS (Osborne et al., 2022). However, the fact that butanol is a flammable liquid, strongly hinders the operation of this type of instrument aboard passenger aircraft. Instead, the application of water-based CPCs is highly advisable, mainly under consideration of flight security aspects.

This study is part of the development of a new air quality package for IAGOS, in response to these flight safety aspects. It

comprises the measurements of the particle size distribution in the diameter range from 125 nm to 4μm using a modified Portable Optical Particle Spectrometer (POPS, (Gao et al., 2016) originally developed by NOAA, of the particle extinction coefficients at different wavelengths as well as the $NO_2$ concentration using four Cavity Attenuated Phase Shift (CAPS, Aerodyne Research Inc., Billerica, MA, USA) instruments, and finally of the total particle number-concentration measured by the water-based MAGIC 210-LP CPC characterised in this work.

The new water-based condensation particle counter (MAGIC 210-LP) for low-pressure applications characterized in this study was recently introduced into the market by Aerosol Dynamics Inc. and is based on the standard MAGIC CPC, which contains a pre-humidifier, where the aerosol sampling flow is guided to a continuous wet wick with different temperature zones. Starting with the cold conditioner region, then the warm initiator and a cold moderator zone before finally passing the optics head (Hering et al., 2019). The MAGIC 210-LP CPC was subjected to counting efficiency experiments for a broad pressure

range and different types of test aerosol particles. The experiments were part of the qualification of the individual components of the IAGOS Air Quality Package under development.

## 2 Methods

A schematic of the experimental set-up is shown in Figure 1. To provide a steady and constant particle production in size distribution and number concentration, a constant output atomizer (Model 3076, TSI Inc., Shoreview, MN, USA) was used,

which nebulizes a constant stream of an ammonium sulphate (AS) solution (Liu and Pui, 1975); (TSI Inc. Model 3076 Manual). After the aerosol flow passes through a diffusion dryer tube, the relative humidity reaches levels below 5%. It follows a charging process by passing through a radioactive Am-241 source and the classification in a monodisperse aerosol takes place by a Vienna-type Differential Mobility Analyzer (DMA, Model M-DMA 55-U, Grimm Aerosol Technik GmbH & Co. KG, Ainring, Germany). This aerosol enters the low-pressure zone by passing through a critical orifice. The aerosol is diluted

within the mixing chamber. The pressure is controlled there as well by a LabVIEW program through multiple mass flow controllers with a PID approach. Furthermore, the relative humidity is actively controlled by adding a stable humidified air



flow into the system through the mixing chamber, which is limited to approximately 30% relative humidity. After passing the mixing chamber, the aerosol flow is provided to the measuring instruments using individual isokinetic, iso-axial samplers located in the centre of the sample line. Here, a Sky-CPC 5.411 (Grimm) was used as a well-characterized butanol

condensation particle counter (Bundke et al. 2015). An aerosol electrometer was used as a traceable reference instrument for particle counting measurements (FCE, Model 5.705, Grimm). The instrument of interest was the newly developed Moderated Aerosol Growth with Internal Water Cycling CPC (MAGIC 210-LP, Aerosol Dynamics Inc, Berkeley CA, USA). For the fresh flame soot measurements, the nebulizer as well as the dehydration tube were replaced by a Miniature Inverted Flame Soot Generator (Argonaut Scientific Corp., Edmonton, AB, Canada). A description with greater detail is provided in

prior studies (Bundke et al., 2015; Bischof, 2022).

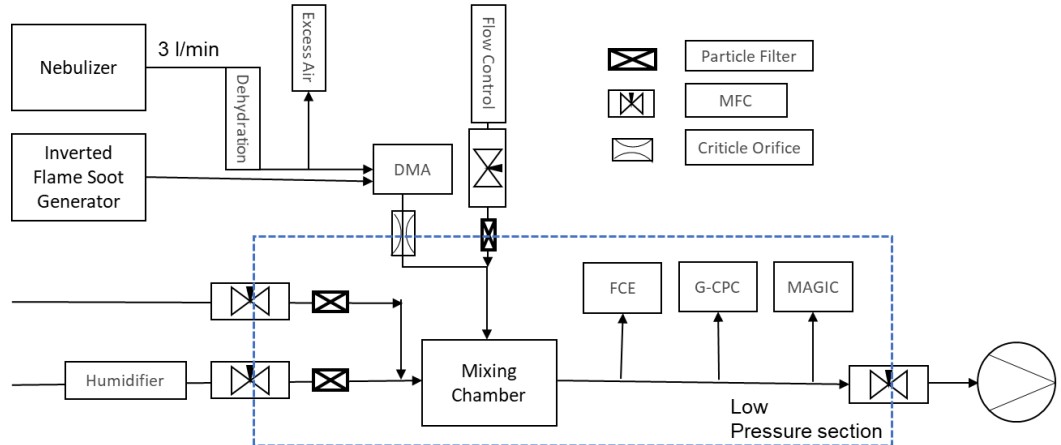

**Figure 1. Flow schematic of the laboratory set up for the low-pressure characterization with two aerosol sources. The inline pressure is controlled via mass flow controllers (MFC); the aerosol size classification is ensured with a differential mobility analyser (DMA) and the**
**faraday cup electrometer (FCE) functions as a reference instrument for particle counting).**

The DMA was operated stepwise for 30 seconds for each voltage level corresponding to different particle sizes starting at an upper limit of 140 nm and going down to 2.5 nm. To avoid transition effects and to achieve an equally distributed aerosol inside all measuring instruments, the first 15 seconds for each particle size setting of the DMA were excluded from the

dataset.

The inverted flame soot generator was operated with an oxidation-air-to-propane ratio of 7.5 L/min air to 0.0625 L/min propane. This ensures stable aerosol production with low organic carbon soot (Bischof et al., 2019; Kazemimanesh et al., 2018).


## 2.3 Data analysis procedure

A major issue for the measurement of nanometre-sized particles using a DMA is the presence of multiple charged particles. These are erroneously selected by the DMA because they have identical electrical mobility as singly charged particles but are larger in size. This effect leads to a notable difference in the counting rate between a condensation particle counter and an



aerosol electrometer. To address this artefact, the correction scheme and routine shown in Figure 3 were used, which was

introduced by Bundke (Bundke et al., 2015; Bischof, 2022).

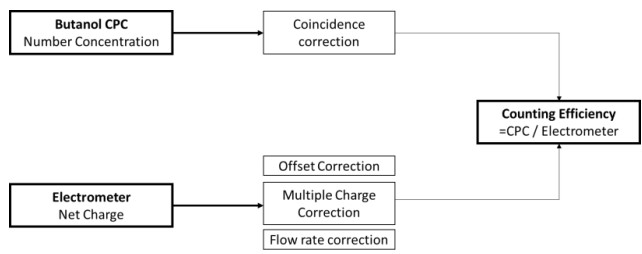

**Figure 2. Flowchart of the data inversion procedure for particle concentration determination.**


The Multiple charge correction can be expressed by

$N_{FEC}^* = \xi(D_p)N_{FCE}$ (Bundke et al., 2015),

as $N_{FEC}^*$ as the corrected Electrometer number concentration and ξ as the calculated correction factor.

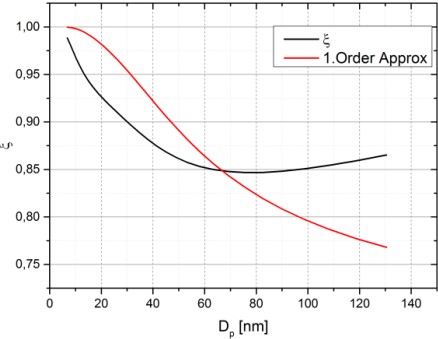

**Figure 3. Multi charge correction function adapted from Bundke et al., (2015).**

Figure 3 shows the multi-charge correction factor ξ(Dp) as a function of the particle diameter. The red line shows the first-

order approximation ignoring the actual size distribution. The first-order approximation deviated significantly (up to 15%)

from the _ξ (Dp) curve. Thus, the actual size distribution measurement needs to be considered.

An exponential fit function introduced by Wiedensohler et al. (1997) (Banse et al., 2001) was used to give a more quantitative

description the efficiency curves.

Equation (1)     $\eta = A - B * (1 + \exp{\frac{(D_p - D_1)}{D_2}})^{-1}$ .

Here, η is the counting efficiency, $D_p$ is the particle size, and A, B, $D_1$, and $D_2$ are fitting parameters.



### 3. Results and discussion

The MAGIC CPC has two variables which are critical for low-pressure measurements. One is the laser power which is adjusted to compensate for variations in droplet size as a function of the operating pressure. The second variable is the detection threshold voltage which was adjusted to compensate for variations in background scattered light (i.e., measured light with
zero particle counts) as the laser power varies. When internal pressure detection is measuring a decreasing pressure, it increases the laser power and decreases the detector threshold, until only the detector threshold is the only limit of signal detection. The MAGIC 210-LP CPC was operated with the temperature settings recommended for low pressures by the manufacturer in the operational manual. The manual for the MAGIC 210-LP states, that the conditioner temperature should be kept at 2°C and the moderator at 4°C for low-pressure operations. The initiator is fixed at 45°C to remain below the boiling
point when operating at pressures as low as 150 hPa. These working points, however, could not be reached if the heatsink exceeds temperatures of 33°C. During a heatwave, it became clear, that the thermoelectrical devices get to their limits. It was then observed, that in case the temperatures of the conditioner and the moderator are about 3 K above their recommended value, the counting efficiency decreases by about 20% from 100% to 80% overall counting efficiency at pressure levels 250 hPa and below. This limitation, however, is solvable by maintaining the ΔT between all temperature zones
of the sections of the growth tube equally.

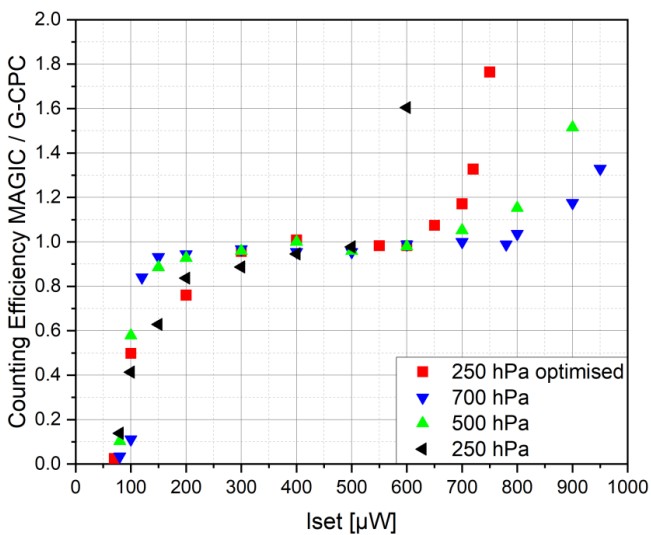

**Figure 4.** Counting Efficiency response for different initial laser power settings (Lset) and pressure levels for 100 nm particles.

As the absolute pressure during operation decreases, also the droplet growth is reduced resulting in smaller droplets to be
counted. This is compensated by adjusting the laser power and the detector treshold. The instrument firmware makes these adjustments automatically based on a lookup table, and the 1-sec averaged reading for the operating pressure. Concentrations are reported with respect to laboratory conditions of 25°C and 1013 hPa.



The following optimization step was applied. The MAGIC 210-LP was designed for operation at pressure levels as low as 300 hPa. We were able to extend the range of operation down to 200 hPa by the adjustment of the initial laser power to different

pressure levels. In Figure 4 the counting efficiency expressed as the number concentration measured by the MAGIC instrument divided by the number concentration measured by the G-CPC instrument is shown at different pressure levels for 100 nm sized particles for a range of initial laser power settings. To compensate the effect that particles grow less efficient at lower pressures the threshold and the laser power are controlled as function of the internal measured pressure. Here the laser power is increased, and the offset is lowed with decreasing pressure values. At 250 hPa the adjusted detector threshold

reaches 0 V and only the constant parameter of the detector offset limits the counted signals. The detector offset adjusts for non-ideal electronics and optics so that the signal without any particles present is at zero volts. The detector threshold is the voltage level that is used to determine if a particle is in the laser beam. In Figure 4 each graph was obtained from measurements at a detector offset setpoint of 250 mV. The only exception is the "250 hPa optimised". To compensate for the increased stray light, the detector offset was increased from 250 mV towards 400 mV. This series of measurements was

conducted with the increased 400 mV detector offset. This method of adjusting the initial laser power setting increased the secure bandwidth of laser power that could be applied, without false counts. A stable counting rate was obtained for the set point at 500 µW for all pressure ranges. For higher pressure levels, the threshold must be set to a value of over 300 mV to compensate for the higher offset of 400 mV at ground pressure levels.

To give an overview of the two particle types we used, the aerosol size distribution for the test aerosol is shown in Figure 5.

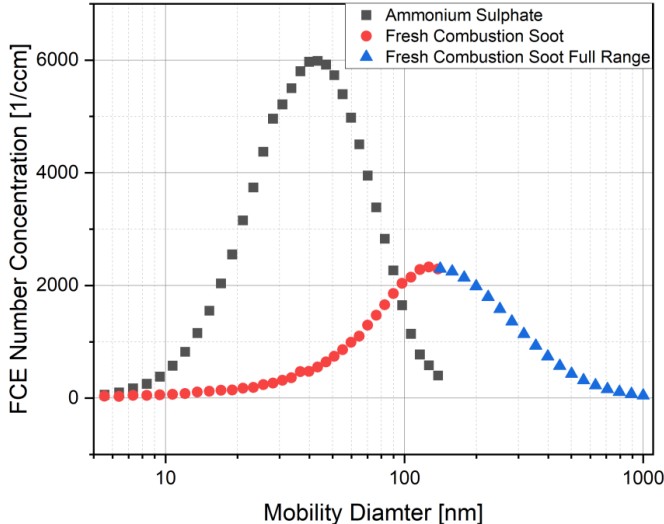

**Figure 5. Particle size distributions were measured by Electrometer and sized by DMA for ammonium sulphate and fresh combustion soot. For this work, the particle mobility sizes were measured to 138 nm, so the size resolution at lower sizes is suitable for the cut-off characterisation. The full particle size distributions are available at (Weber et al., 2022).**


The overall counting efficiency, the cut-off diameter and the linearity of the two condensation particle counters compared to the electrometer used as a reference instrument at different pressure levels was essential to look at for the instrument validation for IAGOS. The measured particle concentrations were compared to the corrected electrometer concentrations.



In Figure 6, the particle size-dependent counting efficiency of the G-CPC and the MAGIC 210-LP concerning the multi-charged

corrected FEC reference measurements are shown. In Figure 7 the scatter plot demonstrates the overall linearity between

the instruments.

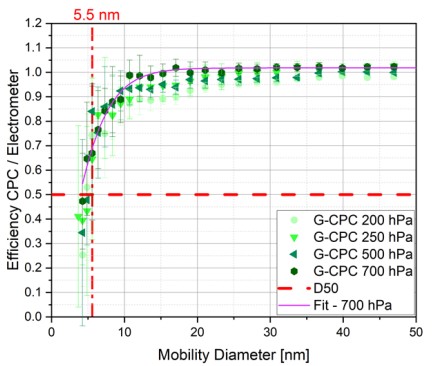 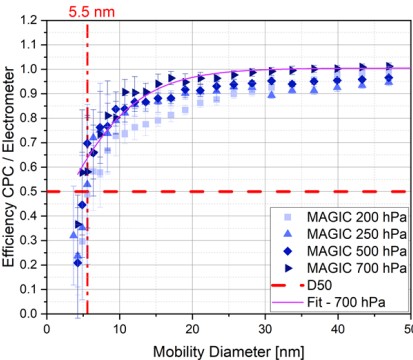

**Figure 6. Compilation of the efficiency ratio curves of the Sky-CPC 5.411 (G-CPC) (left) and the MAGIC 210-LP CPC (right) to the FCE reference - at difference operation pressures as a function of the particle size using ammonium sulphate particles.**


Using ammonium sulphate as a particle material, the instruments respond with an excellent agreement with the FCE

reference instrument, with a slope of 1.0 ±0.05 regardless of the inline pressure. The MAGIC 210-LP and the Sky-CPC scatter

around the 1:1 line, showing counting linearity for the full spectrum of particle concentrations, as illustrated in Figure 7.

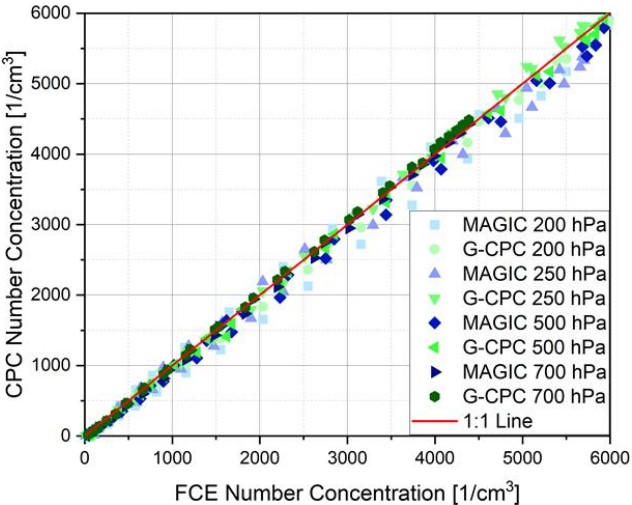

**Figure 7. Comparison of the counting linearity between both CPC types and the Electrometer at different pressure levels for nebulized ammonium sulphate.**

When looking deeper into detail at small particle sizes, both CPCs show a $D_{50}$ cut-off diameter of around 5 nm at all pressure

levels (see Table 1). The reported $D_{50}$ value is in accordance with previous measurements performed with the standard

MAGIC instrument, using ammonium sulphate as aerosol material (Hering et al., 2005). The G-CPC shows no major change



in counting efficiency behaviour when it is operated at reduced pressures. The MAGIC 210-LP counts at least 90% of the particles when compared to the electrometer for pressure levels higher than 250 hPa and particle sizes larger than 30 nm. As the operation pressure reaches 200 hPa, the counting efficiency suffers from a small drop to about 80%, but only for particles smaller than 15 nm. The laser power and detector threshold parameters were chosen to cover all pressures down to 200 hPa.

As a second particle type, we used combustion soot produced utilizing the Miniature Inverted Flame Soot Generator (Bischof et al., 2019). The experimental set-up was therefore adjusted by replacing the nebulizer and its subsequent diffusion drier with the inverted flame soot generator.

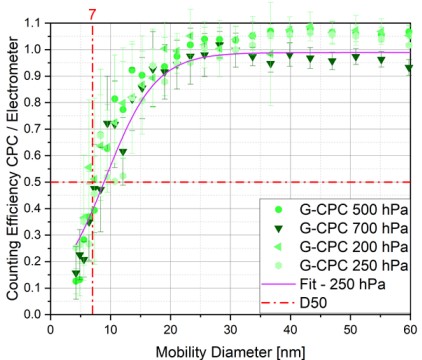 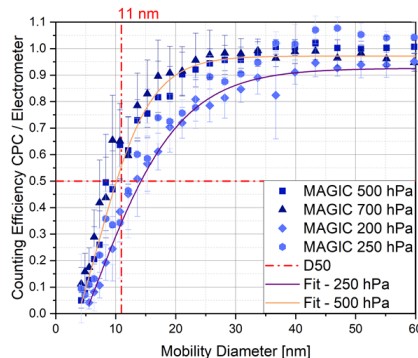

**Figure 8. Comparison of the efficiency ratio curves of the Sky-CPC 5.411 (G-CPC) (left) and the MAGIC 210-LP CPC (right) to the electrometer reference at different operating pressures as a function of the particle size using fresh combustion soot.**

Figure 8 and Figure 9 show the comparison between the Grimm CPC, MAGIC 210-LP and Electrometer for the freshly generated combustion aerosol at different levels of operating pressure. The G-CPC and the MAGIC 210-LP show nearly identical behaviour for counting efficiencies at pressures higher than 250 hPa. For lower pressure, the G-CPC continues to measure with the same efficiency. As soon as an ambient pressure of 200 hPa is reached, the $D_{50}$ cut-off of the MAGIC 210-LP increases to around 15 nm and its $D_{90}$ to about 40 nm.




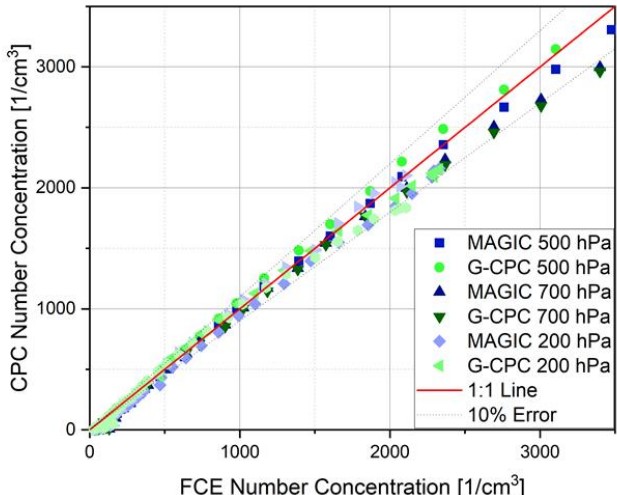

**Figure 9. Comparison of the counting linearity between CPC and Electrometer at different pressure levels for fresh combustion soot.**

As an insoluble hydrophobic substance, fresh combustion soot is not activated for droplet formation inside a CPC as efficiently as hydrophilic substances (Petzold et al., 2005). Therefore, soot particles need to be larger in diameter for nucleus activation than hydrophilic particles, which explains the increase of the $D_{50}$ compared to our ammonium sulphate experiments. For airborne measurements, it is unlikely to encounter fresh combustion soot, but it is a good proxy for non-hydrophilic substances.

Tables 1 and 2 give an overview of the counting linearity of the CPC type instruments with respect to the FCE reference instrument, for both aerosol types. As can be seen, the Person Rate $r^2$ value never drops below 0.99. With respect to the instrument linearity for soot particles, the MAGIC 210-LP as well as the Sky-CPC show up to 15% fewer particles compared to the FCE with increasing total number concentrations. This might be an artefact caused by the multiple charge correction underestimating multiply charged particles. A possible reason for this is that the multiple charge correction is based on the size distribution measurement. For soot, only a part of the full-size distribution is taken into account using the m-DMA, so that the log-normal fit applied to the measurement data might underestimate the right tail of the distribution. This results in an underestimation of multiply charged particles. We aimed for a high size resolution for particle sizes smaller than 30 nm for a precise cut-off efficiency determination, this made it necessary to limit the maximum diameter size.

Looking at the $D_{50}$ Value of Tables 1 and 2, both Instruments have the same range of particle diameter of 5 nm for ammonium sulphate, where only 50% particle detection could be archived towards the reference instrument. This agrees with the reported detection limit for both instruments from the manufacturer and publications (Hering et al., 2014; Bischof, 2022).

When looking deeper into the cut-off efficiency measurements, the overall trend for the MAGIC 210-LP becomes obvious. With decreasing pressure, the delta $D_{50}$ to $D_{90}$ is increasing resulting in a less steep decrease in the counting efficiency towards smaller particle sizes. Thus, resulting in a higher uncertainty for measuring the total number concentration involving the Aitken mode for low-pressure surroundings. Whereas the $D_{50}$ value does not change significantly, the $D_{90}$ increases significantly. The overall large delta $D_{50}$ to $D_{90}$ indicates a shallow decrease in counting efficiency over a wide particle size range. Switching the particle type to soot, the detection limits increase because of the activation affinity of this particle type.




**Table 1. Coefficients of the Exponential Fit of the Counting Efficiency Curves for the Sky-CPC 5.411 (G-CPC) and the MAGIC 210-LP CPC and slopes for different line pressure values and ammonium sulphate.**

| Instrument | Line Pressure | Slope | A | B | $D_1$ [nm] | $D_2$ [nm] | Derived $D_{50}$ [nm] | Exp. $D_{50}$ [nm] | Derived $D_{90}$ [nm] |
|---|---|---|---|---|---|---|---|---|---|
| G-CPC | 200 | 1.03±0.02 | 0.96 | 2 | 2.8 | 1.8 | 4.9 | 4.8±0.7 | 9.1 |
| MAGIC 210-LP | 200 | 0.98±0.01 | 1.03 | 1.7 | 0.1 | 8.1 | 6.8 | 5.5±0.8 | 20.9 |
| G-CPC | 250 | 1.02±0.01 | 1.0 | 2 | 1.7 | 2.6 | 4.6 | 5.5±0.8 | 9.4 |
| MAGIC 210-LP | 250 | 0.95±0.02 | 0.97 | 2 | 1.6 | 3.4 | 5.5 | 5.5±0.8 | 12.9 |
| G-CPC | 500 | 1.00±0.01 | 0.98 | 2 | 3.0 | 1.4 | 4.6 | 5.5±0.8 | 7.5 |
| MAGIC 210-LP | 500 | 0.97±0.01 | 0.95 | 2 | 2.5 | 2.2 | 5.2 | 5.5±0.8 | 10.6 |
| G-CPC | 700 | 1.02±0.02 | 1.0 | 2 | 1.5 | 2.5 | 4.2 | 4.3±0.7 | 8.9 |
| MAGIC 210-LP | 700 | 1.01±0.01 | 1.0 | 2 | 0.1 | 4.1 | 4.6 | 4.3±0.7 | 12.3 |

**Table 2. Coefficients of the Exponential Fit of the Counting Efficiency Curves for the Sky-CPC 5.411 (G-CPC) and the MAGIC 210-LP CPC for different line pressure values and fresh combustion soot.**

| Instrument | Line Pressure | Slope | A | Bb | $D_1$ [nm] | $D_2$ [nm] | Derived $D_{50}$ [nm] | Exp. $D_{50}$ [nm] | Derived $D_{90}$ [nm] |
|---|---|---|---|---|---|---|---|---|---|
| G-CPC | 700 | 0.84±0.01 | 0.92 | 1.2 | 5.9 | 3.2 | 8.0 | 7±1.0 | 19.0 |
| MAGIC 210-LP | 700 | 0.86±0.01 | 0.93 | 1.4 | 5.9 | 3.6 | 8.7 | 9±1.3 | 19.7 |
| G-CPC | 500 | 0.98±0.01 | 1.02 | 2 | 3.6 | 3.8 | 7.6 | 8±1.0 | 14.2 |
| MAGIC 210-LP | 500 | 0.95±0.01 | 0.97 | 2 | 3.6 | 5.2 | 9.6 | 9±1.3 | 21.0 |
| G-CPC | 250 | 0.95±0.01 | 0.99 | 0.94 | 9.2 | 4.2 | 8.6 | 7±1.0 | 19.1 |
| MAGIC 210-LP | 250 | 1.03±0.01 | 1.03 | 2 | 3.3 | 8.9 | 11.9 | 11±1.4 | 27.6 |
| G-CPC | 200 | 0.95±0.01 | 1.01 | 1.03 | 6.6 | 4.2 | 6.7 | 5.5±0.8 | 15.9 |
| MAGIC 210-LP | 200 | 0.94±0.01 | 0.93 | 2 | 3.8 | 7.6 | 13.4 | 13±1.5 | 35.7 |

Analyzing the behaviour of the fitting parameter A, which can be compared to the slopes of the regression lines, since it represents the plateau of the fit function, and the derived parameter $D_{50}$ of the fitting function in Tables 1 and 2, no clear

trend is visible. The values of $D_{50}$, deduced from the fitting curves are close to 5 nm for both condensation particle counters and all pressure levels in case of ammonium sulphate (Table 1) and fit to the experimental data. For fresh combustion soot (Table 2), $D_{50}$ values for the G-CPC instrument are slightly larger at a value of 8 nm, while for the MAGIC 210-LP the





increase in $D_{50}$ compared to ammonium sulphate is more pronounced. Overall, the agreement between values derived directly from the experiment and values deduced from the fitting procedure is high.

At lower pressures, the particle counting efficiency drops for small particle sizes, however, does not impact the quality of the measurements when using a sampling line of more than 1 meter length, as will be the case in applications aboard passenger aircraft equipped with IAGOS instruments. Here 50% (85%) of 5 nm (13 nm) particles will penetrate to the instrument ($P\_150$ hPa, $T\_293$ K, 2.4 L/min total flow) (Bundke, 2015). In such a set-up, particles smaller than 13 nm in diameter will be removed by diffusion during the sampling process. Yet, the overall uncertainties must be determined by modelling the

instrument responses of MAGIC 210-LP and G-CPC for different aerosol size distributions, mainly with and without a nucleation mode, for IAGOS – characteristic sample line lengths

**Conclusions and recommendations**

The MAGIC 210-LP CPC was recently introduced as a new water-based CPC with excellent overall performance compared to a standard Butanol CPC. In this work, we characterised a modified "LP" model of that water-based CPC design for flight

altitude pressure levels as low as 200 hPa. We recommend, testing each unit for low-pressure applications and adjusting the manufacturer settings when operating at pressures lower than 500 hPa if necessary. When operating above this pressure level, the factory settings were satisfactory. We were able to have a look at 5 units. Critical for a high counting efficiency are the laser power, detector offset and detector threshold.  It is noted that since this study, the manufacturer has modified the firmware and design of the MAGIC 210-LP to improve the performance at high altitudes and to better accommodate the

automatic adjustments in the laser and detector settings with operating pressure.

The MAGIC 210-LP operates without loss in performance at all pressure levels tested and reports reliable particle concentrations with overall detection efficiencies close to 100%. Its well-engineered water recycling mechanism makes the instrument attractive for long-term operation in harsh conditions with no or only very limited opportunities for instrument access and maintenance. To evaluate the instrument performance, and in particular, the counting efficiency, as a function

of the aerosol type and pressure, an aerosol electrometer and a butanol condensation particle counter was used as established reference instruments. For ammonium sulphate particles, the MAGIC 210-LP CPC shows excellent stability of the $D_{50}$ cut-off diameter and overall linearity with an $r^2$ of 0.99. Approved by experimental data and an exponential fitting function, the overall counting efficiency reaches 100% for pressure levels higher than 200 hPa and particles larger than 30 nm, regardless of the particle type.  However, at 200 hPa the counting efficiency for particles smaller than 30 nm drops

notably to 90% compared to the electrometer or the butanol CPC. When the MAGIC 210-LP is exposed to a hydrophobic and insoluble particle type like fresh combustion soot, the water condensation particle counter shows similar behaviour for almost all particle sizes down to 30 nm for ambient pressure levels down to 250 hPa when comparing the overall linearity stay within 95%. This pressure range covers the operational conditions present during IAGOS aircraft flights. For pressures below 200 hPa, the efficiency of the MAGIC 210-LP can reach 100% linearity towards the reference instrument for a large

particle range.  For particles smaller than 30nm the counting efficiency is lower than 90 %, decreasing to 70% (60%) for 20 nm (15nm) particles.  Because of the reduction of the counting efficiency for particles smaller than 30 nm for operational pressure levels below 250 hPa, the uncertainty of the reported number concentration is enhanced, particularly when sampling an aerosol with a strong nucleation mode, and the lower counting efficiency of the MAGIC 210-LP for smaller particle sizes results in a higher uncertainty of the total particle count.




*Acknowledgements*. Parts of this work were supported by IAGOS-D (Grant Agreement No. 01LK1301A), and HITEC Graduate School for Energy and Climate at Forschungszenrum Juelich.

*Contributions of co-authors.* PW performed all instrument calibrations, the instrumental set-up, and the data analysis. UB and BF designed the LabVIEW environment of the experimental set-up. MB helped during instrument preparations. SS, GL and SH provided technical details of the instrumentation. PW, OB, UB and AP contributed to the manuscript and the interpretation of the results.

*Conflict of interest.* GL and SH are owners of, and SS is an employee of Aerosol Dynamics Inc, which developed and sell the MAGIC 210-LP. The other authors declare that they have no conflict of interest.

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
