# Peer review of "Characterisation of a self-sustained, water-based condensation particle counter for aircraft cruising pressure level operation"

_EGUsphere, 2022_

## Author Comment (AC1)

Review of "Characterization of a self-sustained, water-based condensation particle counter for aircraft cruising pressure level operation" by Weber et al.

The present manuscript deals with the performance of a new commercial condensation particle counter aimed for the automated operation onboard passenger aircraft. In a laboratory simulation of the flight conditions, the performance was characterized. The thorough characterization of such an instrument for standard operational use is an important task, and it may serve as reference for the deployment of this instrument at other locations. The manuscript shows the results of these calibration measurements and discusses deviates as function of thermodynamic conditions. It generally shows the suitability of the instrument for the intended use, onboard the IAGOS container.

The manuscript is suitable for the publication in AMT, but requires some clarification and corrections. A grammar check for punctuation is suggested.

Remarks/Questions

Abstract: It should be mentioned in the abstract that the instrument was modified after this investigation. Also, maybe a recognizable instrument version of MAGIC should be given to avoid misunderstandings of the applicability.

**ANSWER:**

> **We will add the information: We used the MAGIC-210-LP Version. Currently, the manufacturer produces the MAGIC -250-LP as an updated version, which incorporated insights from this study.**

Figure 1: Many arrows are not straight, which is inadequate for a scheme. What is the difference between the line from the top labeled 'flow control' and the line from the left without label above the humidifier? If the butterfly symbolizes a MFC, why is the additional 'flow control' needed? G-CPC should be explained in the caption. Flow rate ranges (and pressure ranges) should be given for all flows, not only for one. Also the pump symbol should be in the legend for sake of completeness. A better match to 'Humidifier' would be 'Dryer' instead of 'Dehydration'. The caption has an unmatched parenthesis.

**ANSWER:**

> **You are right, this graph needed more love, so we added a few things:**

[Figure]

110-125: I can't make much sense of this section. It is too short to give a real explanation of what was done in this previous work of Bundke et al. 2015. And it doesn't make sense showing a curve from the previous work and then stating, that another curve has ben used. Or is there a different physical meaning between xi and eta?

This section should be thoroughly reworked. Either extend it to give a short explanation what was actually done in which step, or remove it, refer to the literature (and in case, state what was different to the previous approach).

**ANSWER:**

**We moved this section to the supplementary and expanded it. By using a diffusion charger, the particles may carry multiple charges passing the DMA and may then be counted multiple times by the FCE. To correct the FCE count, the number concentrations are multiplied by a size-dependent correction factor calculated by using the size distribution measurement.**

148-163: This seems to be one of the key achievements. But we learn here for the first time, that MAGIC is actually not specified for < 300 hPa, which might be on of the motivations of that study. Therefore, the problem should mentioned in the introduction. Also, some more information on the optimization procedure would therefore be useful (plots). E.g., at which laser voltages and which sensor thresholds / offsets the system operated with what efficiency?

**ANSWER:**

**We mentioned in the Conclusion that it is necessary to test a MAGIC-LP 210, when operating at 500 hPa or below. Since the manufacturer settings my not be optimizes for this. The manual states an automatic adjustment for the laser power and the detector characteristics down to 300 hPa. We will mention this in the introduction as well.**

From 155-157 we learn that detector offset and threshold are different properties, but the

expressions are used before. Maybe a sketch of the instrument and its logic would help here following the explanation. Are offset and threshold applied to the same reference potential or do they apply to different part of the electronics?

**ANSWER:**

We will include a more detailed description in the supplementary.

Figure 1 shows an idealized signal from the optics electronics.  The analog signal is compared to the "detector threshold" (normally 250mV) which produces a digital pulse that increments a counter in the microcontroller.

The "baseline voltage", i.e. the signal with no particles present, could be above or below 0 volts due to imperfection in the optics and electronics, as shown in Figure 2.   There is always some stray light that reaches the photo detector, and all operational amplifiers have some non-zero offset.  To compensate, a "detector offset" is add to the analog signal to adjust the baseline voltage to zero.

Since the stray light reaching the photodetector is proportional to laser power, the firmware automatically adjusts both the laser power and detector offset with pressure. The specific relationship between laser power and detector offset are set at the factory and vary from instrument to instrument.

To operated the MAGIC 210-LP at pressures lower than then it was designed for, voltage offset and detector thresholds had to be determined experimentally below 300 hPa.  At 250 hPa, we found that the required laser power was so high that the electronics was incapable of zeroing out the baseline voltage. To compensate the detector threshold was increased above the factory setting of 250mV (figure 3).

Note: due to specifics of the electronics a larger firmware setting for the detector lowers the baseline. Also the digital pulse are 0-5V; the height was reduced in the figures for clarity.

Based on this study, Aerosol Dynamics Inc. has updated their low pressure CPCs to operate down to 200 hPa.

[Figure]

**Figure 1: Ideal signal from one particle passing through the optics detector**

[Figure]

**Figure 2:   Effect of imperfections in optics and electronics on the baseline voltage.**

[Figure]

**Figure 3:   Detector threshold is increased to compensate for inability of the electronics to completely cancel out the baseline signal at lowest pressure.**

165: Without further explanation, Fig. 5 should be in the method section describing the aerosol generation. What means 'the particle mobility sizes were measured to 138 nm'? There is a size distribution displayed – the maximum of the soot distribution? Why is the size resolution as a result suitable for the cut-off characterization? Y axis: In Fig. 3, the symbol was used, but here a description instead of N_FCE. Unify (Applies also to other plots).

**ANSWER:**

**In order to achieve a high resolution at smaller particle sizes, we started at 138 nm. It seems like an odd number, but it is even in the voltage settings. The full range size distribution was measured by the identical measurement set-up but for another study and used here, for the sake of completeness.**

179: Fig. 6: The D50 apparently doesn't match the fit curve. Which data does D50 refer to? Is the fit curve in this case then suitable? Same applies to Fig. 8

**ANSWER:**

**That is my fault. There used to be more fitting curves and I erased those, to make the figures more clear. The D50 matched back then, I will correct it.**

=====================================

Minor remarks/Corrections

39: Reference format

**Thanks**

42: Reference format

**Acknowledged**

44: 'limited': Chose another wording. Being a greenhouse gas might be unfavorable, but it not really a limitation (compared to a flammable material onboard aircraft). And water vapor is a greenhouse gas, too, though of course weaker.

**It has a very high greenhousegas potential. Another point is, that FC43 reaches its best performance at low pressures and is not suitable at ambient pressures.**

46: Doesn't the reference belong to the second statement after it?

**Will be relocated**

70: Reference format

**OK**

75: Remove 'as well'?

**As well will be removed**

101: What happened to 2.1 and 2.2?

**Good Question; I update this**

102: Multiply charged?

**Acknowledged**

103: It is not an error of the DMA, as it simply selects according to charge-to-size ratios (or effective mobility). It's an error of the data interpretation by assuming a unique effective-mobility/size relationship. The effect of course is correctly described, but I suggest a more careful wording.

**You are right**

104: 'This effect...' These different sizes?

**Thanks for the suggestion**

109: Figure 2 is not referenced (or referenced as Figure 3). Figure 2 doesn't add much information over the text. Remove.

**Will be moved and explained in the Supplementary**

112: While for N_FCE is quite clear, what it should be, the symbol is not explained above. This equation has no number, but the next one ha. Why?

**Will be moved and explained in the Supplementary**

121: '_' before Xi

**Will be moved and explained in the Supplementary**

122: Reference format

**Acknowledged**

130: 'pressure detection': barometer / pressure sensor is measuring

**Acknowledged**

131: 'until only the detector threshold is the only limit of signal detection': unclear. Reword.

And why is the threshold decreased, if the laser power increases? One would expect that also the 'background' light intensity would increase, and therefore the threshold should be increased.

**The detector offset adjusts for non-ideal electronics and optics so that the signal without any particles present is at zero volts. The detector offset decreases the baseline level. We rework this and put it in the supplementary. The detector threshold is the voltage level that is used to determine if a particle is in the laser beam.**

143: 'optimised' should be explained in the caption

**Acknowledged, with the best fitting Laser and Detector Settings**

144: 'droplets, which need to be counted.' ?

**Acknowledged**

146: If there is a 1-sec average, what is the actually reading frequency?

**Actually this is not quite right. The absolute pressure is measured once per second, and adjusts the laser power and threshold each second. The averaging time between one second and 30 minutes is for particle concentration.**

146: 1-sec averaged à 1 s average or one-second average

**One-second moot, changing to 1-Hz pressure reading**

146: It seems that [standard] laboratory ?

**?**

171-173: Quirky. Rephrase.

**Acknowledged**

173: corrected -> multiple-charge-corrected

**Acknowledged**

174: concerning -> with respect to

**Acknowledged**

174:remove  multi-charged

**Acknowledged**

175: FEC -> FCE

**Acknowledged**

179: different

**Acknowledged**

181: The material ammonium sulfate should be mentioned in the text before the curves are discussed.

**Acknowledged**

198: dryer

**Acknowledged**

200: Fig. 8: The pressure levels should be sorted ascendingly. D50 is one time after, one time before fits.

**Acknowledged**

210: Fig. 9: The pressure levels should be sorted ascendingly.

**Acknowledged**

218: 'square of the Pearson correlation coefficient' or 'coefficient of determination' – but where is it?

**Acknowledged. As can be seen is very misleading.**

234: affinity -> disinclination / repugnance

**The detection limits increased when we switched to soot, because of its lower activation affinity."**240: Table 2: Bb -> B ?

 **Indeed**

250-252: As long as there is not bypass sampling in used.

**Sure**

253: Bundke et al. 2015

**Acknowledged**

257: Header has no number

**Acknowledged**

260: 'We recommend, testing' remove comma

**Acknowledged**

263: "It is noted that since this study, the manufacturer has modified…" That should maybe be noted with a remark at the according plots, otherwise a reader might overlook that the plots are no longer applicable to the current instrument generation.

**Acknowledged**

265: So are the manufacturer setting acceptable now for this pressure range?

**For the MAGIC-LP 250 Units, yes.**

267: "Its well-engineered water recycling mechanism…" That information is new. Its relevance for the section is unclear.

**It is relevant for the continuous operation on IAGOS aircraft packages.**

266: "operates without loss in performance" The manuscript dealt with the details of exactly this performance loss, so this general statement doesn't seem to be suitable for the conclusion section.

**This effects only particles with a small diameter. Those are effected with line loss anyway. We will state this in the Supplementary**

269: "To evaluate …" from here a summary start, which should be at the beginning of the last section.

**Acknowledged**

276: Solubility is probably not a directly relevant property here.

**Acknowledged**

277-278: "… Its well-engineered water recycling mechanism." Unclear. Rephrase

**Acknowledged**

278: "For pressures below 200 hPa, the efficiency of the MAGIC 210-LP can reach 100% linearity…" No data for this pressure range were shown in the manuscript, so the conclusion is a bit surprising. Or is this referring to pressure altitudes? Avoid 'can'.

**Acknowledged. We meant at 200 hPa**

296: Some of the references don't seem to be managed by a citation system, what would berecommended. Some dois are given as http-reference, some only numerically. Unify.

**Acknowledged**

300: Details of publication missing?

**Indeed**

303: doi is missing a 'w'

**Oh**

**Citation**: https://doi.org/10.5194/egusphere-2022-1244-RC1

---

## Author Comment (AC2)

Review of "Characterization of a self-sustained, water-based condensation particle counter for aircraft cruising pressure level operation" by Weber et al.

The paper details the performance of a COTS water CPC for use on the IAGOS aircraft. Instrument performance was compared to another butanol-based CPC and an electrometer throughout the pressure range of anticipated flight conditions for two different aerosol species.

The manuscript requires some clarification, added details, corrections, and further editing for grammar and punctuation. However, the discussion is suitable for the publication in AMT.

###############################################################################
###########

Remarks/Questions

Abstract:

You say "simulated aircraft operational environment", but no temperature characterization across ambient range. Is instrument sensitive to ambient temperature changes affecting sample temperature, and thus, supersaturation and cut size?

Saying "excellent agreement" between the instruments is misleading when you have performance differences in pressure for soot particles.

**ANSWER:**

**Measurement data from a separate instrument running in an IAGOS package shows, that the temperature in the package is around room temperature (22-27 °C).**

**We will rephrase that statement.**

Figure 1:

Poorly drawn diagram. Uneven spacing, crooked lines, random box sizes, critical orifice gap and protrusion.

**ANSWER:**

**The Figure is updated**

[Figure]

Low pressure section incorrectly defined at Flow Control filter. Flow control and "dry side" of the humidity section are redundant.

With 4 flow controllers, do you have any measure of stability of the system? How steady was the sample flow, pressure, and humidity control?

**ANSWER:**

**The measured standard deviation for 600 s is 0.24 hPa (100s 0.07 hPa) at 200 hPa. The reported flow data shows a standard deviation of 0.0017 l/min for the "main" flow controller**

92-95:

Why constant 30 second steps? What were your statistics? Your particle size distribution concentration is varying >3 orders of magnitude across the size range (fig 5), why hold constant DMA steps. Increase time at small sizes to reduce massive error bars in counting statistics.

**ANSWER:**

**Earlier experiments have shown that this time is sufficient to flush the system. The "error" bars shown in the figures are the sig+ and sig- values. We will mention this in the manuscript and change it to the variance.**

Why is no data shown above 60 nm if upper limit was 140 nm?

**ANSWER:**

**After 60 nm was no gain of knowledge. The Ratio reached its relatively stable plateau and we wanted to show a clear picture of the graphs at the D50 diameter.**

Is 2.5 nm lower limit corrected for diffusion losses changing distribution shape asymmetrically, and shifting peak upwards? Any line loss analysis to approximate what the actual peak was when DMA is set to 2.5 nm?

**ANSWER:**

**The FCE has the roughly same distance (instrument flow is accounted) from the DMA as the CPC. All measurement instruments got the same aerosol particles.**

**The line loss (DMA-> Line + Line -> Instrument) can be viewed here:**

[Figure]

What is your mixing chamber volume and flowrate to show that 15-seconds between samples is enough flush time?  Show flush time is at least 3-5*Tau.

**ANSWER:**

**The Volume of the mixing chamber is approximately 550 ml. The mixing chamber is flushed with 10 l/min. Therefore, after 15sec, the chamber is flushed more than 4 times.  We tested this by putting a filter in between the DMA and the Mixing Chamber and reached Zero particles within this time.**

101-125:

Section labeling?  Whole section needs to be explained more thoroughly and clearly.

How is ksi determined? Is it calculated, determined experimentally?  Where is the equation for it?  How can one reproduce your correction method with the information provided here?

**ANSWER:**

**We moved this section to the supplementary and expanded it. The theory behind it with full explanation is written in Bundke et al. 2015.  Using a diffusion charger, the particles may carry multiple charges passing the DMA and may then be counted multiple times by the FCE. To correct the FCE count, the number concentrations are multiplied by a size-dependent correction factor calculated by using the size distribution measurement.**

Figure 2: What flowrate and offset corrections?  Why and how are they performed? You have them listed in Figure 2 but never address.  Figure 2 does not add value.

**ANSWER:**

**An FCE measures a zero count. Those were subtracted from the reported particle counts. The Flowrate correction is necessary to address the changing mass flow rates at different pressures.**

148-163:

CPC operating parameters should be mentioned earlier (first introduction of the instrument).

This is your first use of the term "offset" without defining what it is or how it differentiates from the detector threshold.

You stated only two parameters are adjusted, laser power and detector threshold. Now you're adjusting the offset too?

Move definition of offset at its first usage, and first introduce it when you're defining what parameters you adjust.

Move last sentence (162-163) to where you're talking about threshold set points. You jump from threshold set, to laser power adjust, back to threshold set.

**ANSWER:**

**This section will be moved into the supplementary. The manufactures produces now an updated version (MAGIC- LP 250).**

**Figure 1 shows an idealized signal from the optics electronics. The analog signal is compared to the "detector threshold" (normally 250mV) which produces a digital pulse that increments a counter in the microcontroller.**

**The "baseline voltage", i.e. the signal with no particles present, could be above or below 0 volts due to imperfection in the optics and electronics, as shown in Figure 2. There is always some stray light that reaches the photo detector, and all operational amplifiers have some non-zero offset. To compensate, a "detector offset" is add to the analog signal to adjust the baseline voltage to zero.**

**Since the stray light reaching the photodetector is proportional to laser power, the firmware automatically adjusts both the laser power and detector offset with pressure. The specific relationship between laser power and detector offset are set at the factory and vary from instrument to instrument.**

**To operated the MAGIC 210-LP at pressures lower than then it was designed for, voltage offset and detector thresholds had to be determined experimentally below 300 hPa. At 250 hPa, we found that the required laser power was so high that the electronics was incapable of zeroing out the baseline voltage. To compensate the detector threshold was increased above the factory setting of 250mV (figure 3).**

**Note: due to specifics of the electronics a larger firmware setting for the detector lowers the baseline. Also the digital pulse are 0-5V; the height was reduced in the figures for clarity.**

**Based on this study, Aerosol Dynamics Inc. has updated their low pressure CPCs to operate down to 200 hPa.**

[Figure]

**Figure 1: Ideal signal from one particle passing through the optics detector**

[Figure]

**Figure 2:   Effect of imperfections in optics and electronics on the baseline voltage.**

[Figure]

**Figure 3: Detector threshold is increased to compensate for inability of the electronics to completely cancel out the baseline signal at lowest pressure.**

165:

State the aerosol types in the text. Also why you used them as your reference aerosols.

**ANSWER:**

**Ammonium sulphate is an omnipresent aerosol in the atmosphere. Fresh combustion soot is interesting, because the MAGIC should be able to measure non-volatile particle matter emissions from aircraft engines while operating on IAGOS.**

Figure 5 and associated text should be in the Methods section.

**ANSWER:**

**Acknowledged**

Since your data is most significant in the 3-10 nm cut size range, use log y-scale so the concentrations used during the tests are more apparent.

**ANSWER:**

**Great suggestion**

Figure 6: No horizontal error bars accounting for DMA transfer function width. What were your DMA flows? Sizing accuracy analysis?

**ANSWER:**

**The DMA sheath flow was set to 6 L/min , whereas the sample flow was 1 l/min. This narrows the horizontal error down to 1/6 of the mobility according to DMA theory.**

222-225:

If log-normal fit is inappropriate, don't use a log-normal fit.  Use the measured size distribution in your calculations.

Error between log-normal fit and measured size distribution affecting your multiple-charge correction can be calculated.  If you're using this as your explanation, prove it.

**ANSWER:**

**We used the measured size distribution for all calculations. We could not measure the complete combustion soot particles size distribution with without changing to the L-DMA; The  approximation could be an issue; nevertheless, it gave consistent results when checked at the lower pressure ranges.**

281-284: This is your first time discussing uncertainty. This should be discussed in detail in the results section, then summarized in conclusions.

**ANSWER:**

**We tried to make a statement, that for the operation on IAGOS, the reported total number concentration might have a higher "dark number" therefore uncertainty compared to butanol CPC.**

**################################################################################################**

Minor remarks/corrections

11: Remove "and more"

**OK**

26: Punctuation

**Thanks**

31-33: nm particles can be detected via charging and electrometer, as you've used.  Suggest changing to "... growing them to optically-detectable droplets..."

**Acknowledged**

38: Replace "by" a photodiode with "with" or "using"

**OK**

51-53: Wordy. "However, butanol's flammability property strongly hinders..." Also, flammability does not hinder the operation, it hinders the desire to operate it.

**We got no permission to operate butanol CPC on a passenger aircraft. The reason was: flammability. We add this phrase: "Because of it's flammability the use of butanol on passenger aircraft requires special permission which we were unable to attain."**

54-59: Unbalanced parentheses and wordy.

**We will rephrase it, by adding semicolons:**

**This study is part of the development of a new air quality package for IAGOS, in response to these flight safety aspects. The package consists of a modified Portable Optical Particle Spectrometer (POPS, (Gao et al., 2016) originally developed by NOAA which measures of the particle size distribution in the diameter range from 125 nm to 4µm; four Cavity Attenuated Phase Shift (CAPS, Aerodyne Research Inc., Billerica, MA, USA) to measure the particle extinction coefficients at different wavelengths as well as the $NO_2$ concentration; and the water-based MAGIC 210-LP CPC characterised in this work to measure the total particle number-concentration.**

60: define "low-pressure" range. Can it be used in a balloon? High-altitude aircraft?

**We tested it down to 200 hPa. I forgot that low-pressure could be relative.**

63-64: Incomplete sentence by itself. No subject.

**Acknowledged**

64-65: define "broad pressure range" and define "aerosol types" and why.

**Acknowledged**

71-73: Avoid using "it". Define. Rework sentence.

**OK**

77: Why is RH controlled to 30%. Explain significance. State where RH and temperature are measured.

**Acknowledged. We wanted to test, if "low" (up to 30%) humidity can have an impact on aerosol activation for particle counters. RH, T , P, are measured at the mixing chamber and in the instrument.**

102: Change to "multiply-charged particles". What do you mean by measurement? DMAs do not measure particle size, they size-select based on particle mobility. The issue is using a mobility-based selector as an equivalent to a size-selector.

**Thanks, I will use this wording**

103: Replace "these" to avoid being ambiguous.  Hyphen singly-charged.

**Acknowledged**

104: "this effect" is ambiguous.

**Acknowledged**

105-106: "this artifact" ambiguous. To address multiply-charged particles biasing the concentration discrepancy...

**Acknowledged**

111: Why is Multiple capitalized?

**Because I made a mistake.**

113: Why is Electrometer capitalized?

**electrometer it is**

119-125: Mixture of fonts, inconsistent throughout text.  Assume document is printed B&W and can't refer to "red line".  Describe what the first order approx. means.

**OK**

123: efficiency of what?

**Counting rate efficiencies compared to electrometer**

130-132: Confusing.

**Will be rephrased, expanded and put into the supplementary**

132-134: Merge sentences.

**OK**

134: Compared to what temperature values at normal operation?

**We will add this information. During normal (ambient, 1000hPa) operation, the conditioner is maintained at 18 K below and the initiator at 17 above, the heat sink temperature, which is typically a few degrees above ambient. The moderator is temperature is normally set as a function of input dew point to minimize water used. Note: if the input and output dewpoint are equal, no water is used in the instrument. Water evaporated in the initiator is condensed in the moderator and flows through the wick back to the moderator. MAGIC is an acronym for Moderated Aerosol Growth with Internal water Cycling). The user has the option of changing these temperature or setting fixed temperatures**

144-147: This section should be merged within the paragraph above

**OK**

149:  Remove: initial.  Replace "to" with "at".

**OK**

150: Missing comma, missing "is"

**Acknowledged**

151: insert "and is shown..."

**Acknowledged**

152: efficiently

**Acknowledged**

153: comma after pressure.  "as a function"

**Acknowledged**

154: lowered

**OK**

157-158: merge sentences

**OK**

158-159: Be specific and mention for the 250 hPa case...

**Acknowledged**

159-160: redundant

**OK**

174: suggest replacing "concerning" to "with reference to"

**OK**

175: Comma after 7.

**Acknowledged**

183: Not necessary to have "as illustrated in Figure 7". You've already stated you're referring to Fig 7. Sentences seem redundant with message.

**ok**

194-195: This sentence seems out of place here... remove?

**We will rephrase it**

197: Explain why the second test aerosol case is necessary. What are you exploring with the second choice of aerosol?

**We used the second type to show the behavior of an aerosol, that does not dissolve in a liquid.**

214: State this earlier in motivating your methods on why you chose this second case.

**OK**

215: Source? There are many flights and missions targeting fresh combustion. On commercial flight, you're flying in a corridor route that follows other aircraft.

**Good Point, I was speaking of the overall likelihood. We rephrase it**

217: Refer back to Eqn 1 to remind the reader. What is derived vs Exp?

**OK. Derived: With the EQ and EXP: read from the data**

218: As can be seen where?

**We rephrase that**

220-221: Are you saying that your method is incorrect?

**We rephrase it**

222: remove "full-size"

**OK**

234: reference.

Table 1 & 2: Why is Table 2 Bb when Table 1 is B? Exp. stands for experimental or exponential? Unclear.

**Acknowledged. It should be B as in the Equation. EXP for experimental. We rephrase it**

243: Refer back to Eqn 1.

**OK**

249: "agreement... is high". Be quantitative.  e.g. "agreement within 10% throughout the range..." "R^2 of ..."

**OK**

253: Formatting

**OK**

260: remove comma

**OK**

261: insert comma after hPa. Change ot "as necessary"

**OK**

262: "We were able to have a look at 5 units."  Relevance?

**This was not an artefact of one unit.**

267: remove "well-engineered".  You're not in marketing.

**OK**

270: change "was" to "were", since you're comparing 2 objects.

**OK**

272: "Approved" is a strange word choice. Revise?

**Acknowledged Changed to "Verified"**

**277-278 Awkward.  Change   to ….**for ambient pressure levels down to 250 hPa the linearity is within 95%

279: Should "below" be "above"?  You didn't test below 200 hPa.

**Acknowledged, Down to**

---

## Author Comment (AC3)

A commercially available water-based CPC (MAGIC CPC ) is tested over a range of pressures to characterize its performance for use on aircraft, specifically IAGOS. The CPC is tested with both hygroscopic and hydrophilic aerosols and compared with a butanol CPC (GRIMM-CPC). The study, once the concerns mentioned below are addressed, represents a substantial contribution to scientific progress in enabling airborne measurement of aerosol concentrations without the use of toxic or high GHG potential working fluids.

The manuscripts describe in detail testing the water-based CPC in comparison to a reference electrometer and a Grimm-CPC over a pressure range 200-700 hPa. While much of the method and experimental set up are well described and justified, there are a number of aspects which need more full explanation or justification:

- Humified air appears to be added after the DMA, changing the humidity from 5 % to 30 %. Do the particles grow after the DMA size selection because of this? What was the residence time between the DMA and the CPCs and electrometer?

  **ANSWER:**

  > **The Humidity is added after the DMA. This should be below the efflorescence. We wanted to show We wanted to test, if "low" humidity can have an impact on aerosol activation for particle counters. The mixing chamber is flushed every 3-4 seconds. Particle grow due to humidification after size selection has no effect on the results.**

- In figure 1, what is flowing in through the two valves in the bottom section, and the flow controller in the top middle? Is it zero-air? Filtered lab air? Something else?

  **ANSWER:**

  > **It is aerosol-filtered air.**

- What are the flow rates and line lengths between the DMA outlet and CPC and electrometer inlets? Were diffusion losses the same in all 3 lines?

  **ANSWER:**

  > **The Diffusion losses are assumed to be the same. The flexible conductive sampling tubing length from the line to the instruments is 25 cm to instrument that drew 0.6 l/min and adjusted proportionally to instruments with a different flow.**

- The authors discuss how the water-based CPCs respond differently to particles depending on their chemical composition, hence testing both soot and ammonium sulphate. It has previously shown that water-based CPC cut-off diameters can be substantially larger for organic aerosols e.g. (Hering et al., 2005) (by one of the co-authors). Why did the authors choose not to examine organics given their atmospheric relevance?

  **ANSWER:**

**We did not have the appropriate set-up for organic aerosols**. **However, pure soot is hydrophobic and ammonium sulfate is highly hydrophilic, atmospheric organic aerosols should be in between. While pure organics can may be poorly detectable with water CPCs contaminates in even slightly aged particles usually make them detectable with water.**

- Section 3 describes how the CPC increases laser power and decreases the detector threshold "until only the detector threshold is the only limit of signal detection" (page 5, line 132) . I think there may be a slight wording issue here, and beyond that, how is it ascertained when this limit is reached?

  **ANSWER:**

  **We will include a more detailed description in the supplementary. Since the stray light reaching the photodetector is proportional to laser power, the firmware automatically adjusts both the laser power and detector offset with pressure. The specific relationship between laser power and detector offset are set at the factory and vary from instrument to instrument. To operated the MAGIC 210-LP at pressures lower than then it was designed for, voltage offset and detector thresholds had to be determined experimentally below 300 hPa. At 250 hPa, we found that the required laser power was so high that the electronics was incapable of zeroing out the baseline voltage. Aerosol Dyanmics has corrected this in the MAGIC-250-LP.**

- Section 3, page 5, line 139 suggests that CPC overheating in hot ambient conditions could be addressed be maintaining a constant delta T between condenser and saturator, instead of trying to keep each at fixed settings. However, in the previous lines the need to keep the initiator below 45C and the condenser between 2 and 4C because of manufacturer specifications. Do these limits not mean that a constant delta T can therefore not be maintained?

  **ANSWER:**

  **That is correct during a "heatwave" for the low pressure settings, that's why we increased the temperatures of all stages by 3K and were the behaviour of the instrument did not change. For Example: The Conditioner with 5°C and the Moderator with 7°C. 45C was an overly conservative value originally (and mistakenly) specified by Aerosol Dynamics. The boiling point of water at 150 hPa is 52.5 degC. At 200 hPa it is 59 degC. The difference between the conditioner and initiator set point is what determines the D50.**

  **For Ambient Conditions (Pressures around 1000hPa) the Temperature stages are normally operated not with fixed values but with relative temperature settings.**

- Section 3, page 5, line 146 mentions a look-up table for laser power as a function of ambient pressure. Was this something provided by the manufacturer or produced in this study?

    **ANSWER:**

    > **The manual is provided by the manufacturer. The Look-up table is implemented in the software.**

- Section 3, page 6, line 155 to 164 discussed the need to vary the detector offset setpoint from 250 mV to 400 mV. It is not clear to me whether a single detector offset setpoint works for the full pressure range from ground to 200 hPa. Some clarification needed here.

    **ANSWER:**

    > **It varies with pressure and is determined by firmware. We will expand this section and put it into the supplementary**

- Fig 4 – why are there no error bars? The choice of 500 mW laser power seems well justified for the 250-700 hPa pressure range. Why was the lower limit of 200 hPa not tested here? Similarly, what about pressure greater than 700 hPa? Will this setting work to ground level? The authors mention that for 250 hPa and adjustment of the dectector offset was needed from 205 mV to 400 mV. Can this be done automatically in flight? Is there a loss of data while this change is made?

    **ANSWER:**

    > **I simply forgot the error bars. The lower limit of 200 hPa was than tested with the optimized parameters and at all pressure levels up to ground level. This can be done automatically be done by an external command, but with well adjusted parameters, there is no need for that.**

- Fig 5 shows an extended soot size distribution including measurements from another paper by the same 1st Was this size distribution measured with the same experimental set up? If not, what makes it certain that the same size distribution occurred in these experiments as the other Weber et al. 2022 paper?

    **ANSWER:**

    > **It was measured with the same experimental set-up**

- Page 7 Line 181 claims "Using ammonium sulphate as a particle material, the instruments respond with an excellent agreement with the FCE reference instrument, with a slope of 1.0 ±0.05 regardless of the inline pressure." Do the authors mean that a linear fit was performed for each pressure, and all of them had a slope of 1.0 ±0.05 individually? Or was the fit made of the aggregate data over all pressures? It needs to be clearer what was done. If only the aggregate fit was done, the data at higher pressures might make the agreement seem more robust than it really is at low pressure. This graph also lacks uncertainties. Later, around line 220 the authors discuss that both

CPCs see up to 15% fewer particles than the electrometer. This seems to contradict the "excellent agreement with the FCE reference instrument".

**ANSWER:**

> **The 15% fewer particles is exclusively for particles lower than a certain size. We will rephrase the slope-statement.**

- Section 3, page 9, line 220 discusses that the undercounting of the CPCs relative to the electrometer for soot particles may be due to an error in the multiple charge correction because the size distribution was not measured above about 150 nm. But in fig 5, the size distribution is presented using a different study up to 1mm and clearly covering the bulk of the generated mode. Can this extended size distribution not be used to better calculate the multiple charge correction and correct this undercounting?

**ANSWER:**

> **This can be tried. We used the measured size distribution for all calculations. We could not measure the complete combustion soot particles size distribution; The approximation could be an issue; nevertheless, it gave consistent results when checked at lower pressure ranges.**

- In the conclusions, page 11 line 262, it is mentioned that 5 units were examined. This is the first mention of this (would be better in the methods description). It is not clear whether the data presented are from all 5 units, or just 1 CPC. This needs clarification, and the claim that "factory settings for all 5 units" were satisfactory down to 200 mb needs to be supported by data.

**ANSWER:**

> **We will mention this earlier. The data shown are from one unit. All Units must have corrected settings to work down to 200 hPa.**

- Section 3 analyses a number of different aspects of instrument performance, and clarity might be improved by introducing subsections.

**ANSWER:**

**Thanks for the advice**

Many of the conclusions drawn in this manuscript are well justified, but the following do not seem well supported:

- The experiments performed in this study cover the pressure range of 700-200 hPa. Why was 700 hPa chosen as the lower limit? The title suggests that the study only is interested in "cruising pressure levels" in which case the lower cut off of 700 hPa is well justified. This should be made clearer also in the text. The question of instrument performance between 700 hPa and ground seems to be left unanswered.  The

conclusion, page 11 line 278 claims "This pressure range covers the operational conditions present during IAGOS aircraft flights." Do IAGOS flights only use data at pressures lower than 700 hPa?

**ANSWER:**

> **We performed ambient aerosol/pressure measurements and saw no differences as in earlier studies provided by Hering. Because this does not increase the gain of knowledge we did not show these and the 700 hPa measurement data is sufficient as a comparison.**

- Section 3 line 205 claims fig 8 shows the counting efficiencies of the MAGIC CPC and GRIMM CPC are nearly identical for pressure above 250 hPa. This is hard to see from the graphs, and from the fits shown, they seem to have different cut off diameters, so then the behaviour is quite different. I'm not sure what was meant here, and it may be that different plots are needed to show it well.

**ANSWER:**

> **We will rephrase it, to state that 700 and 500 hPa data is meant here.**

- Figure 8 – it is very hard to distinguish the multiple shades of green or blue from each other in these plots, both for the points and the error bars. It makes it hard to determine if the conclusions drawn e.g. about the stability of the G-CPC efficiency curve over the examined pressure range, are valid. The order of the legends makes the plots very confusing to read. The D50 lines are very confusing. They appear to not pass through any of the presented fits. How are they determined? And why do they not pass through the fit lines?

**ANSWER:**

> **That is my fault. There used to be more fitting curves and I erased those, to make the figures clearer. The D50 matched back then, I will correct it.**

- Fig 9 appears to show a trend with ambient pressure of the CPC number concentrations to that measured by the electrometer, but this is obscured by the choice of a linear scale, the hard-to-distinguish colours, the lack of error bars and the confusing ordering of the legend. Until this is corrected, it is not possible to determine whether the claims made around line 205 are justified.

**ANSWER:**

> **We will be more colour friendly and add some. Error bars will be added as well.**

- Page 11 line 249 "Overall, the agreement between values derived directly from the experiment and values deduced from the fitting procedure is high." What objective metric is this based on?

**ANSWER:**

> **The difference between those values is within the uncertainty (which is one half width of the electrostatic mobility)**

- Conclusions, page 11 line 265 "the manufacturer has modified the firmware and design of the MAGIC 210-LP to improve the performance at high altitudes and to better accommodate the automatic adjustments in the laser and detector settings with operating pressure." It is unclear what aspects of "performance" have been enhanced since this study was made, and also which of the laser power and offsets that were tested and adjusted in this study are now different in the instrument settings. The whole concept of what works with the CPC settings and what needs adjusting by the user, and what is adjusted automatically vs needing to be adjusted by the user as the pressure changes needs to be much more clearly addressed.

    **ANSWER:**

    > **To operated the MAGIC 210-LP at pressures lower than then it was designed for, voltage offset and detector thresholds had to be determined experimentally Based on this study, Aerosol Dynamics Inc. has updated their low pressure CPCs to operate down to 200 hPa.**

- Conclusions, page 11 line 258 "excellent overall performance compared to a standard butanol CPC" need to be more specific. The shift in D50 with hygroscopicity should be mentioned, also the shifting D90 with changing pressure.

    **ANSWER:**

    **We will rephrase this accordingly**

Some of the terminology, wording and presentation used is confusing to the reader, and needs adjusting:

- Abstract, line 21 "the D50 cut-off diameter did not differ significantly for particle sizes around 10 nm" – does this mean the D50 cut-off diameter was stable at around 10 nm over the tested pressure range?

    **ANSWER:**

    > **The D50 cut-off diameter did not differ significantly for particle sizes around 10 nm at all pressure ranges.**

    > **We wanted to set the focus on the D90 parameter.**

- Section 3, line 182 references the "Sky CPC", which is otherwise referred to as the G-CPC. It would be clearer to use consistent terminology.

    **ANSWER:**

    > **Acknowledged**

- Introduction, line 26 "adverse effects that particles can have on climate change" seems a bit confused, suggest revision. It makes more sense with reference to air quality, which is listed later.

  **ANSWER:**

  **We will change that order**

- Introduction, line 54 "It comprises", unclear if referring to this study or IAGOS. Suggest it is better to be specific than let the reader infer from the following text.

  **ANSWER:**

  **It should refer to the new IAGOS package. We will rephrase "it"**

- Section 3 line 154: "lowed"?

  **ANSWER:**

  **lowered**

- Section 3, line 207 describes the increase in CPC cut off for soot particles as occurring "as soon as an ambient pressure of 200 hPa is reached", this makes it sound like a sudden transition, whereas in reality it will be a gradual shift with decreasing pressure. The text should be changed to more accurately reflect this.

  **ANSWER:**

  **Thanks for the advice**

- Page 9 line 226, the sentence "Looking at the D50 Value of Tables 1 and 2, both Instruments have the same range of particle diameter of 5 nm for ammonium sulphate" is unclear and needs revision

  **ANSWER:**

  **Acknowledged**

- Tables 1 and 2 – what are A, B and Bb? These do not seem to be defined anywhere.

  **ANSWER:**

  **Those are the Fitting Parameters mentioned in Equation 1 (Bb is a spelling mistake)**

- Conclusions, page 11 line 259 "LP" acronym needs definition, and why is it only introduced here and not earlier?

  **ANSWER:**

  **MAGIC-LP (Low Pressure). We will rephrase it.**

Overall relevant work seems well referenced. I suggest the following additions/corrections:

- Introduction, line 42 – the more relevant reference is again Williamson et al. (2018), which describes the referenced flourinert based CPC in detail, as opposed to a combined payload including that CPC that is described in the current reference.

  **ANSWER:**

  **Will be included**

References:

Hering, S. V., Stolzenburg, M. R., Quant, F. R., Oberreit, D. R., & Keady, P. B. (2005). A Laminar-Flow, Water-Based Condensation Particle Counter (WCPC). *Aerosol Science and Technology*, *39*(7), 659-672. https://doi.org/10.1080/02786820500182123

Williamson, C., Kupc, A., Wilson, J., Gesler, D. W., Reeves, J. M., Erdesz, F., McLaughlin, R., & Brock, C. A. (2018). Fast time response measurements of particle size distributions in the 3–60 nm size range with the nucleation mode aerosol size spectrometer. *Atmos. Meas. Tech.*, *11*(6), 3491-3509. https://doi.org/10.5194/amt-11-3491-2018

---

## Author Response (AR2)

**Public justification (visible to the public if the article is accepted and published)**:
This manuscript contains and important and well-conducted evaluation of the performance of water-based condensation particle counter (CPC) at low pressure. This CPC will be part of an instrument package on the IAGOS measurement suite flying on commercial aircraft. The IAGOS data will be broadly used to improve understanding of aerosol particles in the free troposphere, and is unique in its temporal and spatial coverage. As such, this work is eminently appropriate for publication in AMT, and it is important that the instrument's performance be well documented.

The referee reports of the original manuscript all expressed concerns regarding the clarity and presentation quality of the results. The authors have moved and expanded some technical aspects of changes to the instrument's behavior at low pressure to the supplemental information, retaining a brief description of this topic in the main text.

While the manuscript is improved, it still suffers from a lack of clarity, typographical errors, and some technical issues. While many of these were pointed out by the referees and corrected by the authors, many remain. Because of this, the manuscript is not yet ready for publication.

I point out some of remaining issues below.

1) Typographical errors. There are a number of mis-spellings, incomplete sentences, and other errors in the manuscript. A thorough copy editing job is essential, and an English spell checker must be run to identify some of these errors.

2) Clarity. There are several sentences that are not clear. Some examples (but not all of the cases):
a) Lines 21-22 say, "the D50 cut-off diameter did not differ significantly for particle sizes around 10 nm, whereas. . . .". I think the meaning is, "the D50 cut-off diameter of ~10 nm did not vary substantially as a function of pressure, whereas the D90 cut-off diameter. . . ." (D90 is not defined, by the way.)
     **You Are Right! We have clarified the points in the text.**
b) Line 45, "only reach best performance at low pressure levels". I think the meaning is that butanol CPCs perform best at higher pressures, closer to sea level.
     **This was originally aimed for fluorinert. We changed the text accordingly.**
c) Lines 84-85. "The diffusion losses are assumed to similar for all instruments." Besides missing the "be", this sentence implies an assumption, but in the next sentence we learn that the tubing lengths to the various instruments were adjusted inversely with flow rate, so that diffusion losses were the same for each instrument; no assumption needed.
     **We deleted "assumed"**
d) Lines 118-122. This is a non-sequitur; a statement that is not in the context of the preceding narrative. An exponential fit function? To what? How is the Wiedensohler function related to Equation 1? This is very confusing. I believe the first fit function is for the steady-state charge distribution, but this is not at all clear in the text.
     **EQ1 is a parametric exponential function, that is used to fit the cut-off curves. We have claryfied the points in the text**
e) Lines 208-210. The first part of this sentence says that airborne measurements are not likely to encounter fresh soot, then the second part describes cases where airborne measurements will encounter fresh soot.
     **The overall likelihood to find FRESH soot at crusing altitude (11-13km) is low. But we have observed and identified fresh aircraft plumes onboard our CARIBIC Aricraft by parallel CPC and NOy measurement. These aircraft plumes are visible in time series as 1-2 sec peaks.. We have clarified this in the text.**
f) Lines 154-155. If I didn't understand how DMAs operate, I would have no idea what is being discussed here. Please elaborate that you chose to operate the DMA at flow conditions that would

allow selection of particles as small as 3.5 nm; as a result, the maximum size that could be selected was 138 nm.

**We add the flow condition with 6 L/min sheat flow and the 8.8 cm tube length. We have clarified this in the text.**

3) Scientific/Technical issues. There are some scientific/technical aspects of the manuscript that could be improved, including:

a) Please use a logarithmic axis for all the plots using diameter on an axis. This would allow the greater data resolution at smaller diameters to be clearly seen, and most aerosol parameters naturally scale logarithmically.

**We changed those in the manuscript**

b) The function used to fit to the detection efficiency curves does not describe the data very well in Fig. 4b. Have you considered trying the error function, which is the integral of the lognormal distribution and thus physically based rather than purely parametric? A good reference describing this is Deshler et al., JGR, 2019, doi:10.1029/2018JD029558.

**To be consistent with prior studies, that characterised Butanol CPC instruments, we use this exponential function by Wiedensohler. We redid the fitting function**

c) Does Fig. 5 include data from all diameters, or is the variation in measured concentration for one diameter? Does this range of concentrations encompass the likely range to be observed by the IAGOS package (i.e., from prior measurements in the free troposphere)? I'm curious if there is a roll-off in detection efficiency at higher concentrations. And might there be a roll-off in D50 as well? Remembering that ambient concentration at high altitude will be less than sea-level values, did your measurements cover the expected range? If not, that might be a topic for future studies.

**Yes, Fig. 5 include data from all Diameters.**

**Observation from Butanol CPC instrument during CARIBIC measurements rarely exceeded 6000 cm$^{-3}$ particles, therefore we did not observe a roll-off.**

d) In the Conclusions and Recommendations, it is mentioned that "we were able to have a look at 5 units to verify that this was not an artefact of a single unit." This is extremely important, because the data from multiple instruments on multiple aircraft will likely be combined into a single dataset. If you have quantitative data regarding instrumental variability, please show it, even if it is limited. It is very important to understand how large the unit-to-unit variability might be.

**We did not run the instruments in parallel to give a quantitative statement for the cut-off diameters. This, however, will be done in future studies, with the goal of SOP (Standard operation procedures).**

e) The discussion of diffusion losses of particles is still muddled. Please clearly indicate whether the fall-off in detection efficiency with decreasing diameter at low pressure is due to diffusion losses within the instrument, or if it is more likely caused by pressure-dependent changes in detection due to smaller pulses. I also suggest including a figure showing the calculated diffusion losses in the IAGOs configuration using the one meter tubing length mentioned in the manuscript. You could calculate the losses at 300, 250, and 200 hPa over the entire detected size range. This would be very informative for researchers who will use these data in the future.

**We will include a Figure for 200 hPA and 1000 hPa and the equations in the SI. We used Hinds, W. C., Aerosol Technology: Properties, Behavior, and Measurement of Airborne Particles. Wiley: 1999.**

**22. Baron, P. A.; Willeke, K., Aerosol measurement: principles, techniques, and applications. 2001.**

[Figure]

Aerosol sampling is an important issue, when aerosol measurements claim to cover a total range of aerosol properties and size ranges. Particles can be lost in the sampling line by impaction, sedimentation or diffusion. For very small particles, the diffusion is the most important mechanism for particle losses. Those can be illustrated by diffusion losses in a cylindrical tube under laminar flow. The first thing to recognize is the Stokes`s Law. This fundamental force F describes the total resisting force a spherical particle experience by moving through a medium.

$$F = 3\pi\eta v d$$

With $\eta$ is the viscosity of the medium, v the velocity of the particle and d the diameter of the particle. The Cunningham correction Cc factor has to correct an important assumption of this force. The assumption of the Stokes`s Law is, that the relative velocity of the gas at the surface of the sphere is zero. This assumption becomes an issue for small particles at the range of the mean free path.

$$F = \frac{3\pi\eta v d}{C_c}$$

The Cunningham correction factor can be calculated using the mean free path $\chi$ by

$$C_c = 1 + \frac{\chi}{d}[2.514 + 0.8\exp\left(-0.39\frac{d}{\chi}\right)]$$

Then the diffusion coefficient $\xi$ for laminar flow can be calculated considering the temperature T by

$$\xi = \frac{1.38 \cdot 10^{-23}[\frac{J}{K}] \cdot T \cdot C_c}{3\pi\eta d}$$

With this the dimensionless diffusion parameter $\mu$ can be calculated with the length of the tube l and the air flow rate f.

$$\mu = \xi\frac{l}{f}$$

Finally, the loss fraction can be described by

$$Loss = 0.819e^{-11.5\mu} + 0.0975e^{-70.1\mu} + 0.0325e^{-179\mu}$$

**The final line loss is the product of two settings. Here, the main aerosol sampling line with a flow rate of 4 l/min for the final instrument package and a length of 1 meter till the splitting point, where the MAGIC-LP has a flow rate of 0.3 l/min and 30 cm distance.**

f) What is the volume of the mixing chamber and its e-folding flushing time for the flows used? This information was suggested by one of the referees.

   **The volume of the mixing chamber is 500 ml with a flow rate of 10 l/min. This leads to a flushing time of roughly 3 sec and an e-folding time of 1.8 sec for 63%**

g) Lines 127-129. The detector offset appears here without explanation Perhaps say, "when the internal instrument pressure decreases, laser power is automatically increased. As a result, increasing background scattered laser light effectively increases the baseline voltage of the detector, causing the particle pulse detector threshold to. . . ." And I stop the sentence here because I don't understand what these sentences are trying to say. Please make it very clear what is going on with this detection circuitry, and refer to the SI for more detail. I still don't understand if the baseline change with laser power is a deliberate, controlled shift, or if it is just the result of more scattered light.

   **We changed the text by: The first variable is the laser power which is adjusted to compensate for variations in droplet size as a function of the operating pressure. With decreasing pressure levels, droplet growth is affected, making the droplets smaller. The laser power automatically increases as the internal instrument pressure decreases to compensate for the smaller droplet size. The second variable is the detection threshold voltage which is adjusted to compensate for variations in background scattered light (i.e., measured light with zero particle counts) as the laser power varies, since the background light scattering on molecules increases with increasing laser power […]The manufacturer's settings were not optimized for operating pressure down to 200 hPa. For 250 hPa, we found, that the required laser power was increased so high by the firmware to compensate for the smaller particle sizes, thus the electronics could not determine the baseline voltage correctly. By adjusting the values for the laser power, detector threshold and offset, we solved this issue thus, the MAGIC LP-210 is now able to operate even below 250 hPa. The new settings are applicable for the complete pressure range without change. Further explanation on this is given in the SI.**

4) Copy editing. There are some fairly egregious errors that should not have made it into a submitted manuscript.
a) Figure numbers have changed, but were not updated in the text.
   **We changed those in the text.**
b) The butanol CPC is sometimes referred to as a "Sky-CPC" and sometimes as a "G-CPC". It took me a while to figure out that this was the same instrument. Please be consistent with nomenclature.
   **We will check to be consistent**
c) The atomizer is referred to as a nebulizer in Fig. 1 (which is much improved from the initial submission).
   **We used Nebulizer and Atomizer as synonyms. We check this to be consistent.**
d) What is PID? Also, in Supplemental Information Fig. 1, please make sure all variables are unambiguously defined.
**We update the text with a short explanation: proportional-integral-derivative controller**
e) Line 124. "Low Pressure" has already been defined as part of the instrument name.
   **We deleted this for the Result section, but keep it in the Conclusions**
With these changes made, and a thorough overall job of editing, this manuscript, with its important results, should be suitable for publication in AMT. I look forward to a revised manuscript.

Additional private note (visible to authors and reviewers only):
This manuscript has many technical lapses that should have been fixed prior to submission. I don't expect perfect English usage from a non-native speaker, but I do insist that the meaning of each sentence is clear, which is still not the case. I appreciate the moving of some of the more technical

content to the SI, and the other changes made in response to the initial referee reports (and I thank them all for their detailed and thorough reviews). I strongly recommend that one of the more senior authors take a firm editorial hand with this manuscript and ensure that the intent of each sentence is clearly communicated. With this effort, the paper should soon be ready for publication in AMT.

There is some great information in this manuscript, and I look forward to a revision soon.